# *COL6A1* Promotes Milk Production and Fat Synthesis Through the PI3K-Akt/Insulin/AMPK/PPAR Signaling Pathways in Dairy Cattle

**DOI:** 10.3390/ijms26052255

**Published:** 2025-03-03

**Authors:** Bo Han, Shan Lin, Wen Ye, Ao Chen, Yanan Liu, Dongxiao Sun

**Affiliations:** Key Laboratory of Animal Genetics, Breeding and Reproduction of Ministry of Agriculture and Rural Affairs, National Engineering Laboratory for Animal Breeding, State Key Laboratory of Animal Biotech Breeding, Department of Animal Genetics and Breeding, College of Animal Science and Technology, China Agricultural University, Beijing 100193, China; bohan@cau.edu.cn (B.H.); 202161000080@jmu.edu.cn (S.L.); s20233040757@cau.edu.cn (W.Y.); capudrooks@foxmail.com (A.C.); yananliu1102@163.com (Y.L.)

**Keywords:** *COL6A1*, milk production, milk fat synthesis, dairy cattle

## Abstract

Exploring functional genes/sites and the molecular regulatory mechanisms underlying milk production traits in dairy cattle is crucial for improving the development of the dairy industry and human health. In our previous work, the gene collagen type VI alpha 1 (*COL6A1*) was found to be involved in milk fat metabolism from liver transcriptome data across various lactation periods of cows. Through the integration of Cattle QTLdb, FarmGTEx and qPCR data, the *COL6A1* gene was found to be located within known quantitative trait loci (QTLs), adjacent to single-nucleotide polymorphisms (SNPs) associated with milk traits, and highly expressed in the mammary gland. After employing RNA interference technology, cell function and phenotype tests in bovine mammary epithelial cells revealed that the *COL6A1* gene accelerated cell proliferation, cell cycle progression, and the synthesis of lipids and triglycerides by regulating the PI3K-Akt, insulin, AMPK, and PPAR signaling pathways. Notably, 22 SNPs within *COL6A1* had potential breeding value because they were significantly associated with milk production traits, especially with milk fat. In summary, our findings demonstrate that the *COL6A1* gene promotes milk production and fat synthesis via the PI3K-Akt/insulin/AMPK/PPAR signaling pathways, providing valuable genetic information for molecular breeding programs for dairy cattle.

## 1. Introduction

Milk is rich in various nutrients, including protein, fat, vitamins, and minerals, which can enhance human immunity, promote bone development, and play important roles in preventing chronic diseases [1]. The use of selection methods and techniques to improve the content and composition of milk plays a crucial role in the healthy development of the dairy industry and human health. Since 2009, the adoption of genomic selection (GS) has revolutionized dairy cattle breeding [2,3,4,5], and led to significant changes in the worldwide dairy industry. GS is based on genome-wide high-density markers (mainly SNPs) used to evaluate individual breeding value and select genetically superior individuals [2]. Zhang et al. demonstrated that the accuracies of genomic prediction for milk yield and milk fat percentage were improved by incorporating significant SNP loci associated with target traits in Holstein bulls [6]. Similarly, Brondum et al. and de las Heras Sodana et al. added QTL/SNP loci significantly associated with target traits to the 54K chip genotypes, and the results revealed that the prediction accuracy improved (0.05~5%) for milk production traits of Holstein bulls and carcass traits in Hanwoo beef cattle [7,8]. Therefore, identifying key functional genes/sites and the underlying genetic architecture for milk traits is highly important for dairy cattle breeding.

Since Georges et al. first reported the localization of quantitative trait loci (QTLs) in Holstein cows [9], many QTLs and genetic associations for milk production traits have been discovered. As of 23 December 2024, the Cattle QTL Database (Cattle QTLdb) has released 8809, 25,317, and 46,558 loci for milk yield, milk protein and milk fat traits, respectively (https://www.animalgenome.org/cgi-bin/QTLdb/BT/index, accessed on 23 December 2024). However, the major genes affecting milk traits in dairy cows include only the *DGAT1* and *GHR* genes [10,11,12]. The rapid development of high-throughput sequencing technology provides enormous opportunities for life science research. In our previous research, liver transcription patterns across dry, early lactation and peak lactation periods of Holstein cows were constructed via RNA sequencing (RNA-seq); 12 functional genes for milk fat metabolism were identified, including collagen type VI alpha 1 (*COL6A1*) [13]. COL6A1 is a member of a protein superfamily that plays a role in maintaining the integrity of various tissues, and is a major structural component of microfibrils. Studies have shown that collagen VI is an important component of the extracellular matrix of the breast [14,15]. *COL6A1* participates in Fzd7-Wnt5b signaling, inducing the occurrence and metastasis of breast tumors [15]. However, at present, there is little research on the impacts of this gene on milk traits in dairy cows.

Hence, the purpose of this study was to systematically investigate the regulatory mechanism of the *COL6A1* gene underlying milk trait formation in bovine mammary epithelial cells and evaluate its genetic impacts on milk yield and composition traits in a large population of dairy cows to provide molecular information for the genomic selection breeding of dairy cattle.

## 2. Results

### 2.1. COL6A1 Was Linked to QTLs/SNPs Related to Milk Production Traits

Based on Cattle QTLdb, the *COL6A1* gene (chr.1:145668730-145687976) was located on two QTLs (QTL_1513 and QTL_11321) associated with milk fat yield in dairy cattle; in addition, the SNP rs110959995, related to milk fat percentage and milk protein percentage, was located on intron 33 of this gene, and three SNPs, rs110020706, rs43251562, and rs109680710, associated with milk fat percentage, protein yield, or protein percentage, were 44.333 kb, 471.513 kb, and 497.473 kb away from the 3′ end of this gene, respectively (Figure 1; Appendix A).

### 2.2. COL6A1 Was Highly Expressed in Mammary Gland Tissue

We obtained a total of 1714 RNA-seq data from 27 different tissues of Holstein cows from the FarmGTEx database (https://www.farmgtex.org, accessed on 23 December 2024) [16]. It was found that the *COL6A1* gene had high expression levels in the mammary gland, ovary, uterus, and adipose tissues (Figure 2a; Appendix A); using qPCR, we also found that the *COL6A1* gene was highly expressed in the mammary glands of lactating Holstein cows (Figure 2b).

### 2.3. Cell Proliferation and the Cell Cycle Were Accelerated by COL6A1

To investigate the function of the *COL6A1* gene, we used siRNAs to interfere with *COL6A1* gene expression. In MAC-T cells, siRNA 416 had a highly significant interference effect on the *COL6A1* gene, with an interference efficiency of 68.3% to 75.1% at the mRNA level (Figure 3a; *p* < 0.01). Western blot analysis further confirmed that siRNA 416 downregulated COL6A1 expression (Figure 3b).

#### 2.3.1. Cell Proliferation

The CCK-8 assay results revealed that the cell proliferation of the *COL6A1* gene interference group (siRNA 416) was significantly lower than that of the control group (NC) at 24 h and 36 h post-transfection (Figure 4a); additionally, in the EdU assay, the number of proliferating cells in the siRNA 416 group was significantly lower than that in the NC group at 36 h (Figure 4b).

#### 2.3.2. Cell Cycle

The percentages of cells in the G0/G1, S and M/G2 phases in the interference group were 62.58%, 13.22%, and 5.10%, respectively, and those in the corresponding phases in the control group were 51.55%, 19.45%, and 12.17%, respectively. The proportion of cells in the G0/G1 phase in the interference group was significantly greater than that in the control group (*p* = 0.04), and the proportion of cells in both the S (*p* = 0.03) and M/G2 (*p* = 0.01) phases in the interference group was significantly lower than that in the control group (Figure 5).

#### 2.3.3. Cell Apoptosis

The percentages of living, early apoptotic, late apoptotic, and dead cells in the *COL6A1* gene interference group were 81.10%, 2.50%, 4.23%, and 1.43%, respectively, whereas those in the control group were 79.68%, 3.15%, 4.23%, and 0.55%, respectively. There was no significant difference in the number of apoptotic cells between the two groups (Figure 6; early apoptosis: *p* = 0.22; late apoptosis: *p* > 0.99).

Taken together, these findings suggested that the *COL6A1* gene accelerated cell proliferation and the cell cycle but had no effect on cell apoptosis.

### 2.4. Lipid Synthesis Was Promoted by COL6A1

The cellular lipid and protein phenotypes were also examined. Nile red staining revealed that the red fluorescence intensity was significantly lower in the *COL6A1* gene interference group than in the control group (Figure 7a); additionally, the triglyceride content in the interference group (3.82 ± 0.09 mmol/gprot) was significantly lower than that in the control group (5.80 ± 0.66 mmol/gprot; Figure 7b; *p* < 0.01), indicating that the *COL6A1* gene promoted the synthesis of lipids, especially triglycerides, in MAC-T cells. However, there was no difference in protein concentration between the interference (0.42 ± 0.09 mg/mL) and control (0.44 ± 0.08 mg/mL) groups (Figure 7c,d).

### 2.5. COL6A1 Regulated the Expression of Genes Related to the Cell Cycle and Milk Fat Synthesis

We analyzed the molecular regulatory mechanism of the *COL6A1* gene. This gene was shown to be directly involved in the PI3K-Akt (bta04151) and protein digestion and absorption (bta04974) pathways and connects the AMPK, insulin, and PPAR signaling pathways, in which 28 genes interact, through the core gene *PIK3R1* (Figure 8a). After interfering with *COL6A1*, the expression of the *ITGB1*, *ITGB3*, *ITGB4*, *PIK3R1*, *CCND2*, *CDK4*, *CCNE1*, *SREBF1*, *PRKCI*, *SCD*, *APOA1*, *CD36*, *ACBP*, and *FABP3* genes significantly decreased, whereas *CDKN1B* expression significantly increased (*p* < 0.05). There was no significant difference in the expression of the *ITGA1*, *ITGA2*, *ITGA4*, *ITGA5*, *AKT1*, *AKT2*, *AKT3*, *GSK3B*, *CCND1*, *CDK2*, *CDK6*, *FABP4*, and *FADS1* genes (Figure 8b,c and Appendix A).

Consequently, we constructed a regulatory network for the *COL6A1* gene (Figure 9). In the network, the *COL6A1* gene promotes the expression of *ITGB1*, *ITGB3*, *ITGB4*, *PI3K*, and *SREBP1* through the PI3K-Akt, insulin, AMPK, and PPAR pathways, resulting in the increased expression of *SCD*, *FABP3*, *ACBP*, *APOA1*, and *CD36*, thereby regulating fatty acid and lipid synthesis; modulates fatty acid synthesis through protein digestion and absorption pathways; inhibits *CDKN1B* expression through the PI3K-Akt pathway; and upregulates the expression of the *CDK4*, *CCND2*, and *CCNE1* genes, thereby affecting cell cycle progression (Figure 9 and Appendix A).

### 2.6. COL6A1 Was Associated with Milk Production Traits in Holstein Cattle

Considering the regulatory effect of the *COL6A1* gene on milk production and fat synthesis, we subsequently evaluated its genetic effects on five milk traits in 947 Holstein cows. By resequencing the *COL6A1* gene, we identified 22 SNPs, 5 of which are located in the 5′ flanking region, 15 in introns, and 2 in exon 35 (Appendix A). Association analysis revealed that the 22 SNPs were significantly associated with at least one milk production trait (*p* = 3.20 × 10^−5^~0.04; Table 1); among them, 4 SNPs (rs477773706, rs382853810, rs210433593, and rs442384907) were significantly associated with milk yield (*p* = 3.20 × 10^−5^~0.01); 20 SNPs (rs477773706, rs382853810, rs135913259, rs134137302, rs210433593, rs133101037, rs442384907, rs135906064, rs136643841, rs42427809, rs134989007, rs109837561, rs136361519, rs42427812, rs137589167, rs42427748, rs135517857, rs42427776, rs42427803, and rs132741637) were significantly associated with milk fat yield and/or percentage (*p* = 8.71 × 10^−5^~0.04); and 18 SNPs (rs135862366, rs134299884, rs135913259, rs134137302, rs210433593, rs133101037, rs442384907, rs135906064, rs136643841, rs134989007, rs109837561, rs136361519, rs137589167, rs42427748, rs135517857, rs42427776, rs42427803, and rs132741637) were significantly associated with protein percentage (*p =* 4.90 × 10^−3^~0.04), among which rs210433593 (*p =* 6.32 × 10^−5^) and rs442384907 (*p* = 0.04) were also significantly associated with protein yield. The phenotypic variance ratio (PVR) of six SNPs significantly associated with fat percentage was greater than 1.00% (Table 1). There are 16, 22, and 22 SNPs with significant additive, dominant, and substitution effects on milk traits, respectively (*p* < 0.05; Appendix A). Furthermore, via Haploview, we found that all 22 SNPs formed one haplotype block (D’ = 1; Figure 10), in which nine kinds of haplotypes were inferred with frequencies of 20.9%, 20%, 16.8%, 15.4%, 11.2%, 7%, 3.5%, 2.7%, and 2.5%. The haplotype block was significantly associated with milk yield (*p* < 1.00 × 10^−4^), fat yield (*p* < 1.00 × 10^−4^), fat percentage (*p* = 0.03), and protein yield (*p* < 1.00 × 10^−4^; Table 2). These data demonstrated that the *COL6A1* gene had significant genetic impacts on milk production traits, especially fat traits, in Holstein cattle.

### 2.7. Transcriptional Activity of COL6A1 Was Affected by SNPs

Five of the above twenty-two candidate functional SNPs were located in the upstream regulatory region of the *COL6A1* gene (Table 1 and Appendix A). To explore their influence on gene expression, we used MatInspector to predict the impact of these SNPs on TFBS. It was predicted that allele G of rs134299884 created the binding site (BS) for the transcription factor (TF) PRDM15 (score = 0.90), and allele A created the BS for the TF SMAD3 (score = 0.95). Additionally, allele A of rs135913259 provided a BS for TF AP2 (score = 0.94), TCFAP2E (score = 0.92), and INSM1 (score = 0.96), and G created a BS for TF CMYB (score = 0.99) and RFX5 (score = 0.94; Figure 11a).

To verify the effects of two SNPs, rs134299884 and rs135913259, on *COL6A1* gene expression, we conducted a dual-luciferase assay. As shown in Figure 11, the luciferase activities of the four constructs were significantly greater than those of the empty vector (pGL4.14 + pRL-TK) and the blank control (*p* < 0.01), confirming the transcriptional-regulatory effects of rs134299884 and rs135913259 on the *COL6A1* gene. The relative fluorescence expression levels from high to low were GG, GA, AG, and AA, indicating that the G site of these two SNPs had stronger effects on *COL6A1* gene transcription activation than A, and the G allele of rs134299884 had a stronger effect on gene expression than rs135913259. It can be speculated that these two SNPs may regulate gene expression through binding TFs, and thus affect the formation of milk production traits.

## 3. Discussion

Preliminary transcription data revealed that the *COL6A1* gene is a candidate gene for milk production traits [13]. In addition, the *COL6A1* gene is located on or near to the QTLs or SNPs significantly associated with milk production traits, and is highly expressed in mammary gland tissue in cattle (Holstein [16] and Hereford [17]), indicating that this gene may regulate the lactation function of mammary tissue. In this study, we further validated that the *COL6A1* gene may promote milk production and fat synthesis via the PI3K-Akt/insulin/AMPK/PPAR signaling pathways.

Milk is synthesized and secreted by mammary epithelial cells. Milk production in the mammary gland is determined by the number of secretory mammary epithelial cells and the activity of those cells, and the number of these cells is controlled by the balance between proliferation and apoptosis [18]. This study revealed that the *COL6A1* gene can promote the proliferation and cell cycle of mammary epithelial cells but has no significant effect on cell apoptosis, demonstrating that it promotes milk production.

We observed that after interfering with the expression of the *COL6A1* gene, the expression of the cell cycle-related gene *CDKN1B* was upregulated, whereas the expression of *CCND2*, *CCNE1*, and *CDK4* was downregulated. The cell cycle is the process of cell division and the production of two new daughter cells, which is mediated by cyclin-dependent kinases (CDKs) and their regulatory cyclin subunits [19], and different cyclins bind to different CDKs to regulate events at different stages of the cell cycle [20]. CCND2 and CCNE1 belong to a highly conserved cell cycle protein family, and their members exhibit significant protein abundance cyclicity throughout the entire cell cycle. CDK2 and CDK4 are members of the serine/threonine protein kinase family, and they are especially critical during the G1 to S phase transition. *CDKN1B* encodes a cyclin-dependent kinase inhibitor that prevents the activation of cyclin E-CDK2 or cyclin D-CDK4 complexes and thus controls cell cycle progression at G1. In the normal cell cycle, the E-type cell cycle proteins CCNE1 and CCNE2 bind to *CDK2* to promote G1/S transformation [21]. The downregulation of *CCNE1* expression inhibits cell proliferation and increases the sensitivity of gastric cancer cells to cisplatin [22]. CCND2 couples with CDK4 to initiate DNA synthesis in granulosa cells [23,24], and this process can be suppressed by CDKN1B [25]. Considering that these genes influenced by *COL6A1* expression are closely related to cell cycle progression, we infer that the *COL6A1* gene can inhibit the expression of *CDKN1B* and increase *CDK4*, *CCND2*, and *CCNE1* expression, thereby promoting cell cycle progression and accelerating cell proliferation. Notably, we detected significant genetic associations between the SNPs/haplotype blocks of the *COL6A1* gene and milk yield and composition traits. The positive impact on the milk production phenotype is most likely due to the ability of the *COL6A1* gene to promote cell cycle progression and cell proliferation.

Cow milk contains 3.5% fat, 98% of which is triglycerides. Triglycerides are synthesized by the esterification of a variety of fatty acids, and the endoplasmic reticulum of mammary epithelial cells is the main site of triglyceride synthesis. In this study, we revealed that the *COL6A1* gene connects the insulin, AMPK, and PPAR signaling pathways, which play critical roles in fat and fatty acid synthesis through the PI3K-AKT pathway, thereby leading to the significant upregulation of the downstream genes *SCD*, *FABP3, APOA1*, *CD36*, and *ACBP*, which are related to fat or fatty acid synthesis. The *SCD* gene encodes an enzyme involved in fatty acid biosynthesis, and research has shown that the upregulation of *SCD* expression leads to an increase in the content of saturated fatty acids, including palmitic acid and stearic acid, as well as monounsaturated fatty acid, oleic acid, and palmitoleic acid [26]. *FABP3* was reported to be involved in the milk fat synthesis signaling pathway of dairy cow mammary epithelial cells [27]. APOA1 is a major protein component of high-density lipoprotein and is also present in large triglyceride-rich lipoproteins, very-low-density lipoproteins, and chylomicrons [28,29], which play important roles in lipoprotein metabolism. CD36 binds to long-chain fatty acids and may play a role in fatty acid transport and/or as a regulator of fatty acid transport [30,31]. The *ACBP* gene encodes acyl-CoA-binding protein, a hormone-regulated protein involved in lipid metabolism and a fat-generating factor that triggers food intake and obesity [32]. Hence, we speculate that the *COL6A1* gene promotes the expression of the *SCD*, *APOA1*, *CD36*, and *ACBP* genes through the PI3K-Akt, insulin, AMPK, and PPAR pathways, ultimately promoting the synthesis of fatty acids and lipids.

The cell content test in this study revealed that the *COL6A1* gene had no effect on total protein content, but genetic association analysis revealed a significant correlation between the SNP locus within *COL6A1* and milk protein traits. The possible reason may be the genetic correlations among milk yield and protein and fat contents. Studies have shown that adding functional loci for target traits to GS chips can improve prediction accuracy [6,7,8]. Thus, the significant SNPs associated with milk yield and milk fat traits, identified in this study, could be put into the breeding chip and applied in the genomic selection programs in dairy cattle, thereby increasing the evaluation reliability.

Our study has several limitations, which are the directions of our follow-up research. For example, (1) a larger sample size is needed to verify the genetic impacts of the *COL6A1* gene on milk yield and composition traits, and (2) in-depth functional validation is needed to elucidate the regulatory mechanism of specific loci on trait formation at the cell or organoid level.

## 4. Materials and Methods

### 4.1. Animals and Ethics Approval

The eight tissues, heart, liver, spleen, lung, kidney, mammary gland, lymph, and rumen, were collected from three healthy lactating Chinese Holstein cows for the detection of *COL6A1* gene expression. For genetic association analysis of the *COL6A1* gene, a total of 947 Chinese Holstein cows of 45 sire families from 22 dairy farms were selected, and the corresponding blood and semen samples were collected. Each cow had 3 generations of pedigree information and dairy herd improvement (DHI) records of the first lactation period. The descriptive statistics for the five milk production traits are shown in Appendix A. All the samples and dates were provided by Beijing Sunlon Livestock Development Co., Ltd. (Beijing, China).

All experiments were carried out in accordance with the Guide for the Care and Use of Laboratory Animals and approved by the Institutional Animal Care and Use Committee (IACUC) at China Agricultural University (Beijing, China; approval number: DK996).

### 4.2. Screening of QTLs/SNPs Related to Traits Linked to the COL6A1 Gene

To explore the associations between the *COL6A1* gene and milk production traits, we searched for QTLs/SNPs related to five milk traits (milk yield, milk fat, fat percentage, milk protein, and protein percentage) in the Cattle QTL Database (Cattle QTLdb, https://www.animalgenome.org/cgi-bin/QTLdb/BT/index, accessed on 23 December 2024), with a distance of less than 500 kb [33] from the gene (Gene ID: 511422).

### 4.3. Expression Pattern of the COL6A1 Gene

The expression data for the *COL6A1* gene in Holstein cows, including a total of 1714 RNA-seq samples from various tissues, were obtained from the FarmGTEx database (https://www.farmgtex.org, accessed on 23 December 2024) [16]. The gene expression levels were quantified via transcripts per million (TPM), a method that normalizes read counts by transcript length and the total number of reads per sample to enable accurate comparisons across different tissues.

Furthermore, the expression of the *COL6A1* gene in eight different tissues, namely the heart, liver, spleen, lung, kidney, mammary gland, lymph, and rumen, was assessed via qPCR. Total RNA was extracted using TRIzol reagent (Invitrogen, Carlsbad, CA, USA) and reverse-transcribed to cDNA using a PrimerScript H RT kit (TaKaRa, Dalian, Liaoning, China). The qPCR primers (Appendix A) for *COL6A1* and two reference genes, glyceraldehyde-3-phosphate dehydrogenase (*GAPDH*) and MARVEL domain-containing 1 (*MARVELD1*), were designed with Primer3 (v. 0.4.0; http://bioinfo.ut.ee/primer3-0.4.0/, accessed on 23 December 2024) and synthesized at the Beijing Genomics Institute (BGI, Beijing, China). Then, qPCR was conducted in a LightCycler^®^ 480 II (Roche, Penzberg, Germany) using SYBR green fluorescence (Roche) according to the procedures presented in Appendix A. We performed all the measurements in triplicate, and the relative gene expression levels were normalized to those of the two reference genes via the 2^−ΔΔCt^ method. The data were analyzed using a two-tailed Student’s *t* test, and the differences were considered statistically significant at *p* < 0.05.

### 4.4. COL6A1 RNA Interference in Bovine Mammary Epithelial Cells

Three small interfering RNAs (siRNAs; Appendix A) targeting the *COL6A1* gene were designed and synthesized by GenePharma Biotech (Shanghai, China). Bovine mammary epithelial cells (MAC-T) were grown in Dulbecco’s modified Eagle’s medium (DMEM; Gibco, Life Technologies, Carlsbad, CA, USA) supplemented with 10% fetal bovine serum (FBS; Gibco) under 5% CO_2_ at 37 °C. A total of 8 × 10^5^ cells per well were seeded into 6-well plates. The cells were cotransfected with 5 μL of siRNA/negative control (NC) (20 μM) and 5 μL of Lipofectamine 2000 (Thermo Scientific, Hudson, NH, USA). At 36 h post-transfection, the cells were harvested, and total RNA and protein were isolated. The RNA isolation and qPCR assays were performed as described above. The lower the expression of the *COL6A1* gene after interference, the greater the interference efficiency. The siRNA with the highest transcriptional interference efficiency was screened by protein expression analysis. Protein was isolated via radioimmunoprecipitation assay (RIPA) lysis buffer supplemented with phenylmethanesulfonylfluoride (PMSF). The concentration of total protein was determined via a BCA assay using a total protein assay kit (Nanjing Jiancheng Bioengineering Institute, Nanjing, China). Proteins were separated by 12% SDS-PAGE and then transferred to polyvinylidene difluoride (PVDF) membranes. The membranes were blocked for 1 h and incubated overnight with an anti-COL6A1 antibody (1:1000; Affinity Biosciences, Cincinnati, OH, USA) or an anti-actin antibody (1:1000; abs132001; Absin Bioscience, Shanghai, China). After washing, the membranes were incubated for 1 h with horseradish peroxidase (HRP)-conjugated goat anti-rabbit IgG (1:3000; ZSGB-BIO, Beijing, China). Protein bands were visualized using BeyoECL Plus (Beyotime, Shanghai, China).

### 4.5. Detection of Cell Proliferation, Cell Cycle Progression and Apoptosis

#### 4.5.1. Cell Proliferation

A total of 5 × 10^4^ MAC-T cells per well were seeded into 96-well plates and transfected with siRNA or the NC in sextuplicate. Then, 10 μL of Cell Counting Kit-8 (CCK-8; Dojindo, Shanghai, China) was added to each well at 12 h, 24 h, 36 h, and 48 h post-transfection, and the cells were incubated at 37 °C for 2 h. The absorbance at 450 nm was measured using a microplate spectrophotometer. Additionally, the MAC-T cells were seeded in 12-well plates at a density of 4 × 10^5^ cells/well and cultured for 24 h before transfection with siRNA or NC in triplicate and for 36 h after transfection at 37 °C for the 5-ethynyl-2′-deoxyuridine (EdU) incorporation assay (BeyoClick™ EdU Cell Proliferation Kit with Alexa Fluor 594; Beyotime). The cells were incubated for another 2 h after adding 10 μM EdU to each well, fixed with 4% paraformaldehyde (PFA), incubated for 15 min and permeated with Triton X-100 (Beyotime, Shagnhai, China) for 20 min at room temperature. After washing three times with phosphate-buffered saline (PBS; Gibco), the click reaction mixture was added to each well, and the cells were subsequently incubated for 30 min and stained with Hoechst 33,342 for 10 min at room temperature in the dark.

#### 4.5.2. Cell Cycle

A total of 8 × 10^5^ cells per well were seeded into 6-well plates and transfected with siRNA or NC. Thirty-six hours after transfection, the cells were harvested, fixed with 70% cold ethanol, and stored at 4 °C for at least two hours. Fixed cells were washed with PBS, treated with RNase A (Beyotime), and stained with propidium iodide (PI; Beyotime) for 30 min in the dark. The stained cells were analyzed via flow cytometry (FACSCalibur; BD Biosciences, San Jose, CA, USA). The cell debris and fixation artifacts were gated out, and the cell populations that were at the G0/G1, S and G2/M phases were quantified using Modfit software (Verity Software House, Topsham, ME, USA). Each assay was performed in sextuplicate.

#### 4.5.3. Cell Apoptosis

Cells were seeded and cultured the same as described above for cells used for the cell cycle tests. After the cells were harvested, they were stained with 5 μL of annexin V-FITC (Beyotime) and 5 μL of PI (Beyotime) in the dark for 10 min and then detected by flow cytometry within one hour. FlowJo 10.0.7 software was used to calculate the number of cells. Each assay was performed in quadruplicate.

### 4.6. Lipid Staining and Triglyceride and Total Protein Content Analyses

#### 4.6.1. Staining

Cellular lipids were assessed by Nile red (9-diethylamino-5H-benzo[α] phenoxazine-5-one) staining. A total of 4 × 10^5^ cells were seeded in each well of 12-well plates and transfected with siRNA or NC. At 36 h post-transfection, the cells were washed three times with cold PBS, fixed with 4% PFA for 20 min, and then stained with 500 μL Nile red solution (10 μg/mL; GoldenClone Biotechnology, Beijing, China) in the dark for 15 min at room temperature. Then, the cells were washed again as above and dyed with 500 μL of 4′,6-diamidino-2-phenylindole (DAPI; 100 μg/mL; Thermo Scientific) for 5 min in the dark. The fluorescence indices for Nile red and DAPI were examined with a Nikon 800 Eclipse microscope (Nikon Corporation, Tokyo, Japan) equipped for epifluorescence with a mercury lamp. Each assay was performed in triplicate.

#### 4.6.2. Content Analyses

MAC-T cells were treated as described above; subsequently, the triglyceride concentration in the cells was measured using the glycerol lipase oxidase (GPO-PAP) method with a triglyceride assay kit (Nanjing Jiancheng Bioengineering Institute), where red quinone is formed from triglycerides through cascades of chemical reactions, and the optical spectrum absorbance at 510 nm is directly proportional to the triglyceride content. Similarly, the total protein content of the cells was determined with a total protein assay kit (Nanjing Jiancheng Bioengineering Institute). A standard curve was drawn on the basis of the detection data of gradient-diluted standard samples, and the protein content in each sample was calculated. Each assay was performed in quadruplicate. Differences in contents between knockdown cells and controls were analyzed via a two-tailed Student’s *t* test.

### 4.7. Regulatory Genes in Pathway Analysis and Verification

The KEGG (Kyoto Encyclopedia of Genes and Genomes) PATHWAY database was used for functional enrichment analysis of the *COL6A1* gene, and the interactions between the genes in the regulatory pathways of *COL6A1* were analyzed using Metascape (http://Metascape.org/gp/index.html, accessed on 23 December 2024). Changes in the expression of related genes were assessed via qPCR (Appendix A). The samples used for the qPCR assay were derived from the RNA extracted from the above-described cells with COL6A1 interference and control cells.

### 4.8. SNP Identification and Phenotype-Genotype Association Analysis

#### 4.8.1. SNP Identification and Genotyping

The genomic DNA from the semen of 45 bulls and blood samples from 947 cows was extracted using the salt-out procedure and a TIANamp Blood DNA Kit (Tiangen, Beijing, China), respectively. The quantity and quality of the extracted DNA were determined using a NanoDrop2000 spectrophotometer (Thermo Scientific) and gel electrophoresis. Using Primer3, 22 pairs of primers (Appendix A) were designed to amplify the entire coding region and 2000 bp of the 5′ and 3′ flanking regions of the *COL6A1* gene (NC_037328.1). Then, 1 μL of diluted semen DNA from each bull (50 ng/μL) was mixed for PCR amplification (Appendix A); subsequently, the PCR amplification products were sequenced via Sanger sequencing at the Beijing Genome Institute (BGI), and the sequences were compared with the bovine reference sequence (ARS-UCD1.2) to identify potential SNPs. Furthermore, the identified SNPs in 947 cows were genotyped by genotyping by target sequencing (GBTS) technology by Shijiazhuang Breeding Biotechnology Co. Ltd. (Shijiazhuang, China).

#### 4.8.2. Genetic Association Analysis

The extent of linkage disequilibrium (LD) between the identified SNPs was estimated via Haploview (Broad Institute of MIT and Harvard, Cambridge, MA, USA). Single-locus-based and haplotype-based association analyses were subsequently implemented using the MIXED procedure of SAS software (version 9.4, SAS Institute, Cary, NC, USA):**y** = **µ** + **HYS** + **b** × **M** + **G** + **a** + **e**,(1)
where **y** represents the phenotypic value of each trait (including 305-day milk yield, fat yield, fat percentage, protein yield, and protein percentage) of each cow in the first lactation; **μ** represents the overall mean; **H**, **Y**, and **S** represent the fixed effects of the farm (1–22: 22 farms), calving year (1–4: 2012–2015), and calving season (1: April–May; 2: June–August; 3: September–November; and 4: December–March), respectively; **M** represents the age of calving as a covariant; **b** represents the regression coefficient of the covariant **M**; **G** represents the genotype or haplotype combination effect; **a** represents the individual random additive genetic effect, with a distribution of $N\;\left(0,\;A\delta_{a}^{2}\right)$, where **A** represents a pedigree-based relationship matrix and $\delta_{a}^{2}$ is the additive genetic variance; and **e** represents the random residual, with a distribution of $N\;\left(0,\;I\delta_{e}^{2}\right)$, where the unit matrix is I and the residual variance is $\delta_{e}^{2}$. Bonferroni correction was carried out to adjust for multiple tests, and the significance level was equal to the original *p* value divided by the number of genotype or haplotype combinations. Additionally, the additive (a), dominant (d), and substitution (α) effects were calculated as follows:a = (AA − BB)/2,(2)d = AB − (AA + BB)/2,(3)α = a + d(q − p),(4)
where AA, BB, and AB are the least square means of the milk fat traits in the corresponding genotypes, p is the frequency of allele A, and q is the frequency of allele B.

#### 4.8.3. Proportion of Phenotypic Variance

The effect of an SNP on a specific trait was measured as the proportion of phenotypic variance in the trait explained by the SNP. The proportion of variance explained by an SNP was calculated as follows:phenotypic variance ratio (PVR) = 2pqα^2^/σ_p_^2^,(5)
where p is the allele frequency of the SNP, α is the average effect of gene substitution calculated on the basis of the linear mixed model, and σ_p_^2^ is the estimate of phenotypic variance using the complete DHI data of the Chinese dairy cattle population.

### 4.9. Gene Transcription Activity Analysis

MatInspector software (http://www.genomatix.de/matinspector.html, accessed on 23 December 2024) was used to predict whether the SNPs in the 5′ flanking region of the *COL6A1* gene changed the transcription factor-binding sites (TFBSs; relative score > 0.90). Furthermore, the transcriptional activity of *COL6A1* before and after mutation was verified using a dual-luciferase reporter assay. The fragments containing the binding sites of the transcription factor in the 5′ flanking region of the *COL6A1* gene were synthesized by Genewiz Company (Suzhou, China), cloned, and inserted into the pGL4.14 Luciferase Assay Vector (Promega, Madison, WI, USA). The plasmid constructs were sequenced via Sanger sequencing to confirm the integrity of each insertion and were subsequently purified using an Endo-free Plasmid Maxi Kit (ComWin Biotech, Beijing, China). Human embryonic kidney 293T (HEK293T) cells were used to perform the luciferase reporter assays. Cells were grown in DMEM supplemented with 10% FBS. All the cells were maintained at 37 °C in a humidified incubator containing 5% CO_2_. The cells were seeded in 24-well plates at approximately 2 × 10^5^ cells per well before transfection. For each well, 500 ng of the constructed plasmid was cotransfected with 10 ng of the pRL-TK Renilla luciferase reporter vector (Promega) using Lipofectamine 2000. All the experiments were performed in triplicate. The cells were harvested at 36 h after transfection, and the activities of firefly and Renilla luciferases were measured using a Dual-Luciferase Reporter Assay System (Promega) on a Modulus microplate multimode reader (Biosystems, Sunnyvale, CA, USA). The normalized luciferase data (firefly/Renilla) were used to calculate the average statistics of the replicates.

## 5. Conclusions

In summary, our study showed that the *COL6A1* gene promotes milk production and fat synthesis via the PI3K-Akt/insulin/AMPK/PPAR signaling pathways, and 22 SNPs within the gene have potential breeding value on milk production traits, especially with milk fat, providing valuable genetic information for molecular breeding programs for dairy cattle.

## Figures and Tables

**Figure 1 ijms-26-02255-f001:**
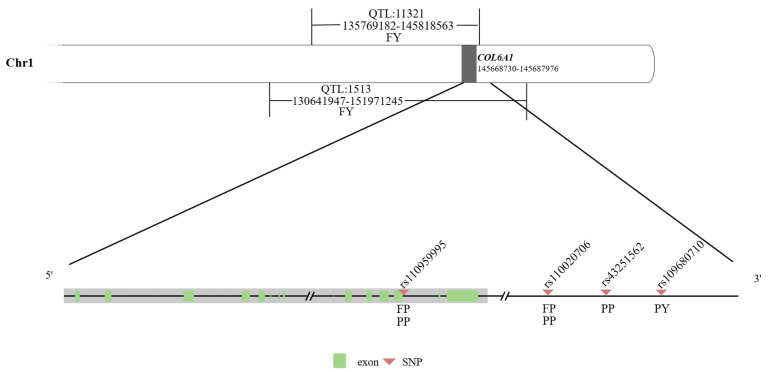
Diagram of the *COL6A1* gene and its nearby QTLs/SNPs associated with milk production traits. FY: fat yield; FP: fat percentage; PY: protein yield; and PP: protein percentage.

**Figure 2 ijms-26-02255-f002:**
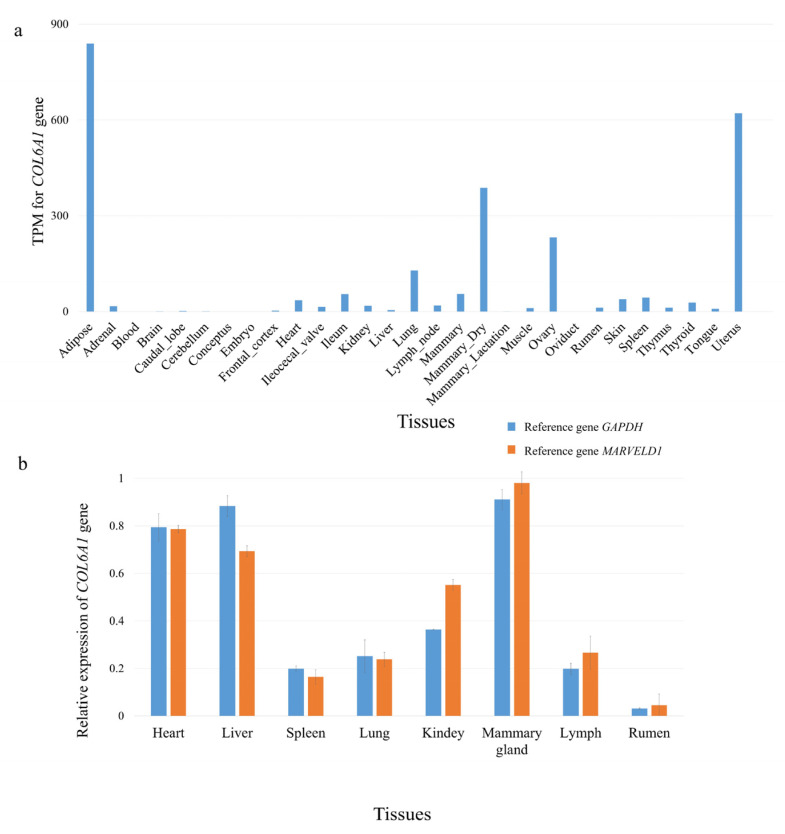
Tissue expression profile of the *COL6A1* gene. (**a**) Expression of the *COL6A1* gene in 27 tissues. The data used for the analysis were obtained from FarmGTEx. (**b**) Relative mRNA expression of *COL6A1* in eight tissues of lactating Holstein cows normalized to that of two reference genes, *GAPDH* and *MARVELD1*, as determined via qPCR.

**Figure 3 ijms-26-02255-f003:**
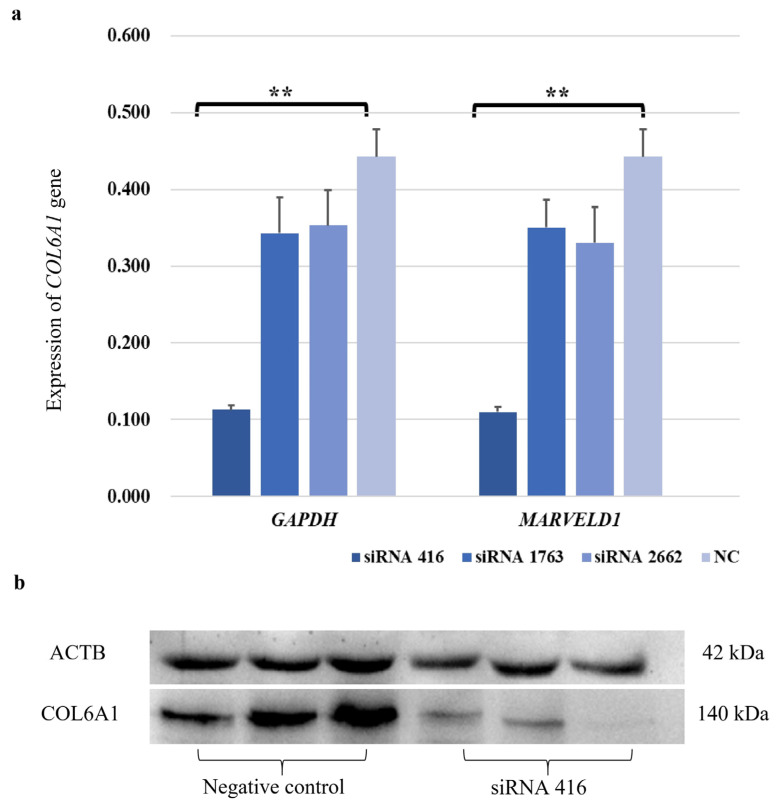
Interference efficiency of siRNA 416 on *COL6A1* gene in mRNA (**a**) and protein (**b**) levels in MAC-T cells. ** *p* < 0.01.

**Figure 4 ijms-26-02255-f004:**
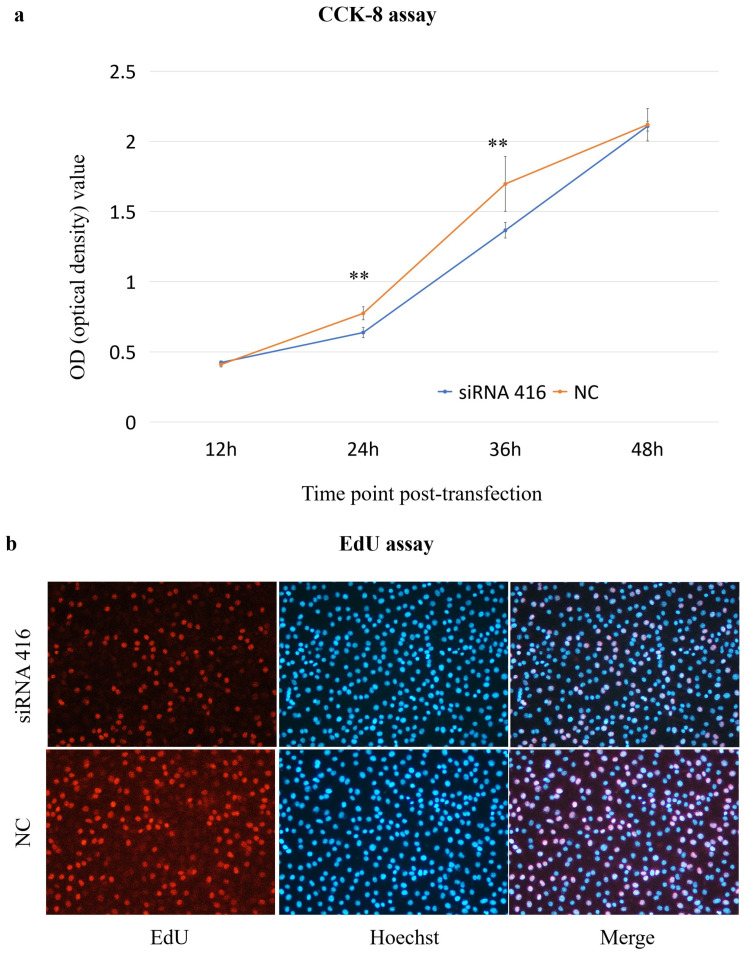
Analysis of cell proliferation ((**a**): CCK-8 assay; (**b**): EdU assay) of MAC-T cells with (siRNA 416) or without (NC) *COL6A1* gene interference. ** *p* < 0.01.

**Figure 5 ijms-26-02255-f005:**
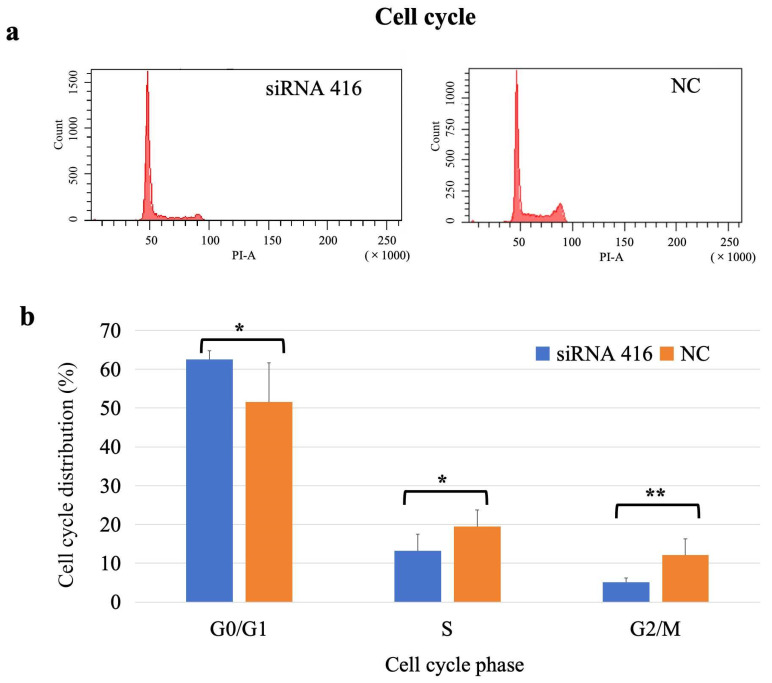
Analysis of cell cycle of MAC-T cells with (siRNA 416) or without (NC) *COL6A1* gene interference. (**a**) Flow cytometry analysis. (**b**) Cell cycle distribution. * *p* < 0.05. ** *p* < 0.01.

**Figure 6 ijms-26-02255-f006:**
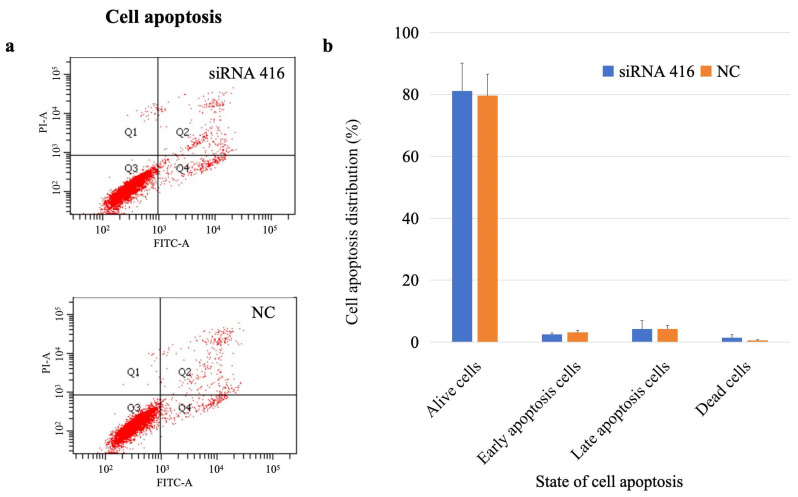
Analysis of cell apoptosis of MAC-T cells with (siRNA 416) or without (NC) *COL6A1* gene interference. (**a**) Flow cytometry analysis. (**b**) Cell apoptosis distribution.

**Figure 7 ijms-26-02255-f007:**
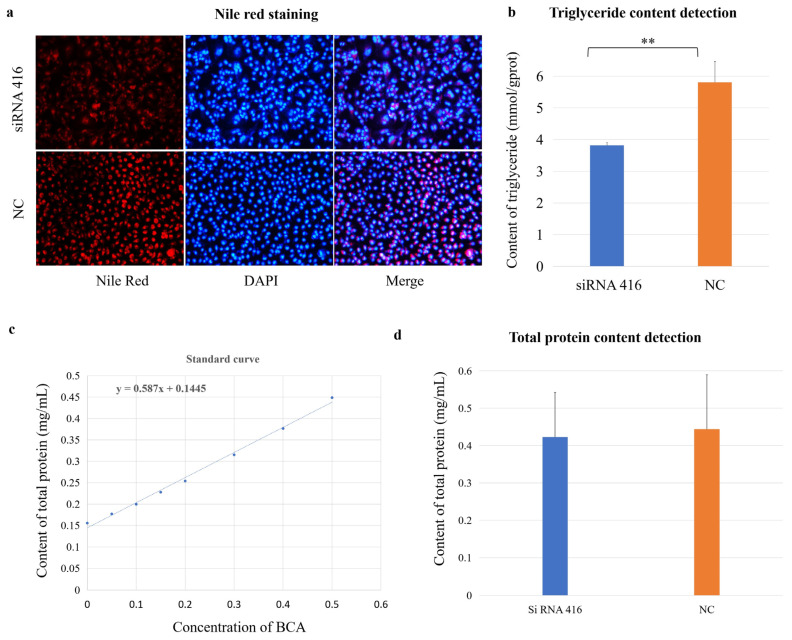
*COL6A1* regulates milk fat synthesis. Nile red staining (**a**), triglyceride (TG) (**b**), and total protein (**c**,**d**) content detection of MAC-T cells with (siRNA 416) or without (NC) *COL6A1* gene interference. ** *p* < 0.01.

**Figure 8 ijms-26-02255-f008:**
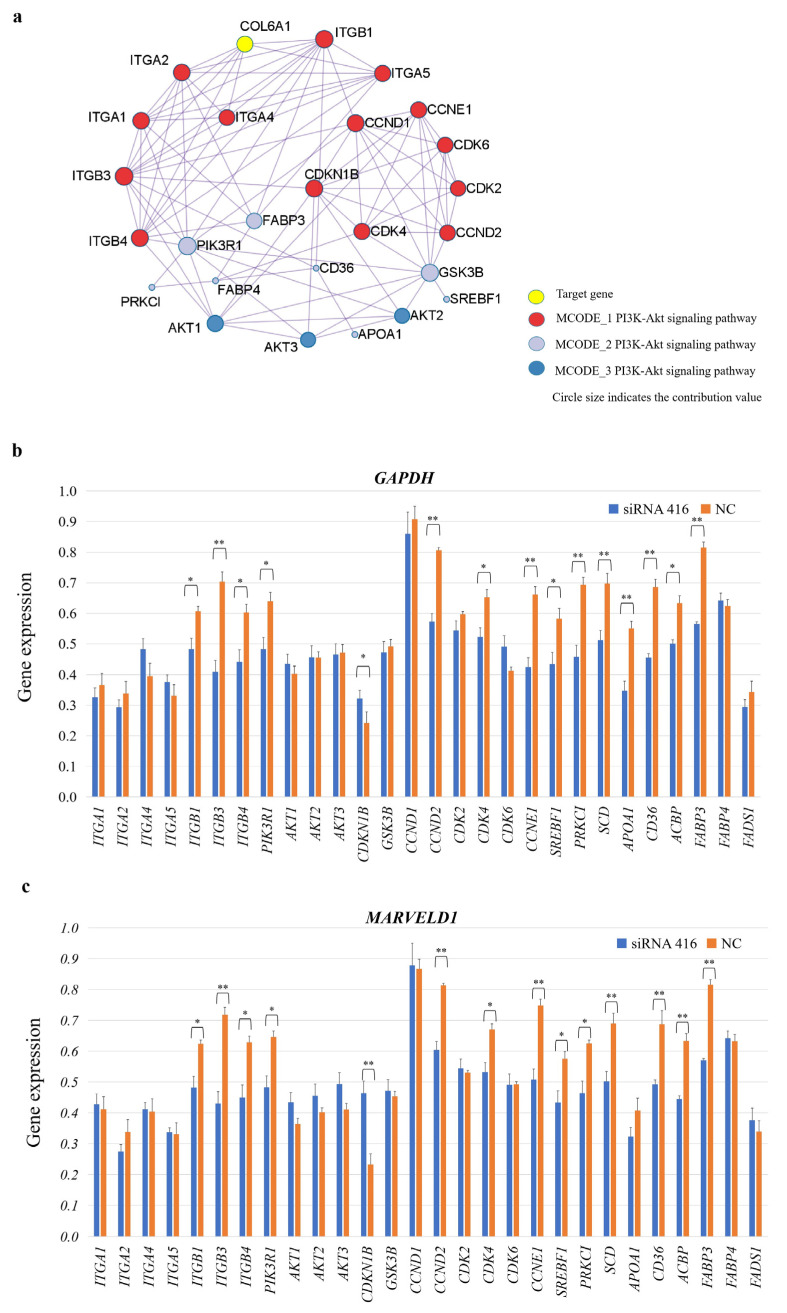
*COL6A1* regulates milk fat synthesis by modulating related gene expressions. (**a**) Regulatory network of the *COL6A1* gene and 28 related genes. (**b**,**c**) Analysis of gene expression levels in related pathways in MAC-T cells with or without *COL6A1* gene interference, normalized by two reference genes, *GAPDH* (**b**) and *MARVELD1* (**c**). * *p* < 0.05. ** *p* < 0.01.

**Figure 9 ijms-26-02255-f009:**
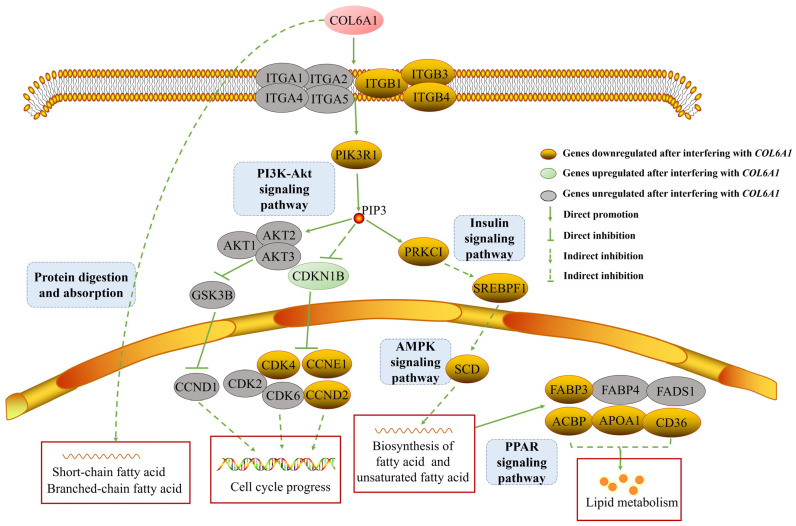
*COL6A1* affects cell cycle and milk fat synthesis by PI3K-Akt, protein digestion and absorption, insulin, AMPK, and PPAR signaling pathways.

**Figure 10 ijms-26-02255-f010:**
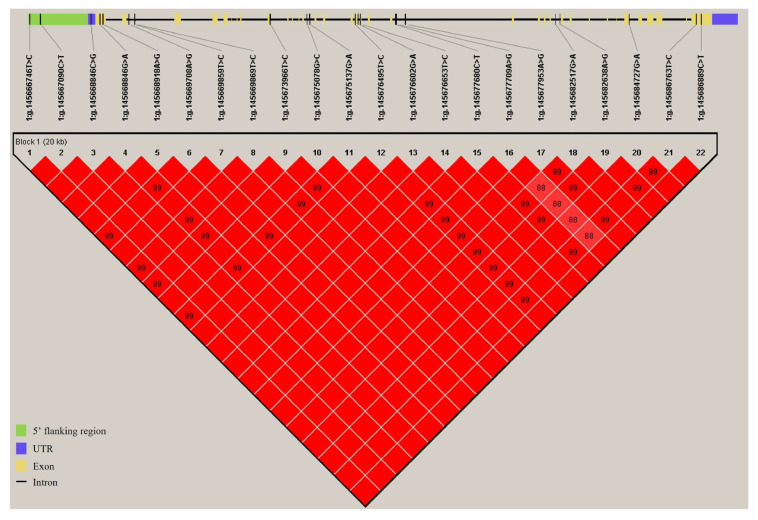
Haplotype block formed by SNPs in *COL6A1*. The blocks indicate haplotype blocks, and the text above the horizontal numbers represents the SNP names. The values in boxes are pairwise SNP correlations (D′), whereas bright red boxes without numbers indicate complete LD (D′ = 1).

**Figure 11 ijms-26-02255-f011:**
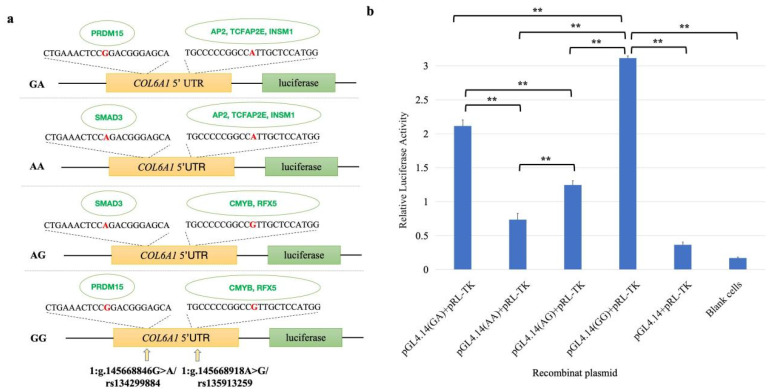
Dual-luciferase activity assay. (**a**) Sketches of recombinant plasmids with rs134299884 (G/A) and rs135913259 (A/G) in the 5′ flanking region of the *COL6A1* gene. The red base is the SNP site. The green ones are transcription factors that can bind to sites. (**b**) Luciferase activity analysis of the recombinant plasmids in HEK 293T cells. PGL4.14 + pRL-TK was the empty vector. ** *p* < 0.01.

**Table 1 ijms-26-02255-t001:** Genetic associations of 22 SNPs in the *COL6A1* gene with milk production traits in Chinese Holstein cows (LSM + SE).

SNP	Genotype (No.)	Milk Yield (kg)	Fat Yield (kg)	Fat Percentage (%)	Protein Yield (kg)	Protein Percentage (%)
rs135862366	CC (50)	10,056.00 ± 117.37	340.65 ± 4.87	3.41 ± 0.05	340.65 ± 4.87	3.00 ^Aa^ ± 0.02
	CT (266)	10,251.00 ± 69.99	337.1 ± 3.06	3.3 ± 0.03	337.1 ± 3.06	2.97 ^b^ ± 0.01
	TT (631)	10,268.00 ± 60.74	339.46 ± 2.71	3.33 ± 0.03	339.46 ± 2.72	2.96 ^Bb^ ± 0.01
	P	0.16	0.52	0.05	0.52	**0.01**
	PVR (%)	0.36	0.05	1.09	0.08	0.84
rs477773706	CC (613)	10,201.00 ^b^ ± 60.41	337.71 ^B^ ± 2.70	3.33 ± 0.03	301.85 ± 1.97	2.97 ± 0.01
	CT (291)	10,326.00 ^ab^ ± 71.08	338.85 ^B^ ± 3.11	3.31 ± 0.03	303.84 ± 2.26	2.95 ± 0.01
	TT (43)	10,530.00 ^a^ ± 127.58	355.27 ^A^ ± 5.28	3.41 ± 0.05	308.28 ± 3.85	2.94 ± 0.02
	P	**0.01**	**1.83 × 10^−3^**	0.09	0.16	0.09
	PVR (%)	0.51	1.51	1.06	0.21	0.07
rs382853810	CC (614)	10,192.00 ^b^ ± 60.40	337.41 ^B^ ± 2.70	3.33 ± 0.03	301.56 ± 1.97	2.97 ± 0.01
	CG (290)	10,349.00 ^a^ ± 71.22	339.59 ^B^ ± 3.11	3.31 ± 0.03	304.58 ± 2.26	2.95 ± 0.01
	GG (43)	10,539.00 ^a^ ± 127.60	355.57 ^A^ ± 5.28	3.41 ± 0.05	308.57 ± 3.84	2.94 ± 0.02
	P	**2.42 × 10^−3^**	**1.32 × 10^−3^**	0.09	0.07	0.10
	PVR (%)	0.49	1.47	1.06	0.20	0.07
rs134299884	AA (50)	10,056.00 ± 117.37	340.65 ± 4.87	3.41 ± 0.05	301.28 ± 3.54	3.00 ^a^ ± 0.02
	AG (266)	10,251.00 ± 70.00	337.10 ± 3.06	3.30 ± 0.03	303.39 ± 2.22	2.97 ^ab^ ± 0.01
	GG (631)	10,268.00 ± 60.74	339.46 ± 2.71	3.33 ± 0.03	302.53 ± 1.97	2.96 ^b^ ± 0.01
	P	0.16	0.52	0.05	0.77	**0.01**
	PVR (%)	0.36	0.05	1.09	0.03	0.84
rs135913259	AA (630)	10,265.00 ± 60.75	339.56 ± 2.72	3.33 ^ab^ ± 0.03	302.46 ± 1.97	2.96 ^Bb^ ± 0.01
	AG (267)	10,258.00 ± 69.96	336.88 ± 3.05	3.30 ^b^ ± 0.03	303.54 ± 2.22	2.96 ^b^ ± 0.01
	GG (50)	10,058.00 ± 117.37	340.61 ± 4.87	3.41 ^a^ ± 0.05	301.31 ± 3.54	3.00 ^a^ ± 0.02
	P	0.17	0.44	**0.04**	0.71	**0.01**
	PVR (%)	0.38	0.05	1.13	0.03	0.86
rs134137302	AA (631)	10,267.00 ± 60.75	339.68 ± 2.72	3.33 ^ab^ ± 0.03	302.52 ± 1.97	2.96 ^Bb^ ± 0.01
	AG (266)	10,253.00 ± 69.97	336.61 ± 3.05	3.30 ^b^ ± 0.03	303.40 ± 2.22	2.96 ^b^ ± 0.01
	GG (50)	10,057.00 ± 117.35	340.57 ± 4.87	3.41 ^a^ ± 0.05	301.28 ± 3.54	3.00 ^Aa^ ± 0.02
	P	0.17	0.35	**0.03**	0.76	**0.01**
	PVR (%)	0.37	0.06	1.15	0.03	0.86
rs210433593	CC (52)	10,287.00 ^AB^ ± 119.30	339.87 ^B^ ± 4.9	3.34 ± 0.05	298.41 ^AB^ ± 3.60	2.91 ^B^ ± 0.02
	CT (293)	10,067.00 ^B^ ± 68.36	332.30 ^B^ ± 2.99	3.32 ± 0.03	297.57 ^B^ ± 2.18	2.97 ^A^ ± 0.01
	TT (602)	10,340.00 ^A^ ± 61.15	342.08 ^A^ ± 2.73	3.33 ± 0.03	305.64 ^A^ ± 1.98	2.96 ^A^ ± 0.01
	P	**3.20 × 10^−5^**	**8.71 × 10^−5^**	0.94	**6.32 × 10^−5^**	**4.90 × 10^−3^**
	PVR (%)	0.20	0.14	0.01	0.01	1.40
rs133101037	CC (50)	10,057.00 ± 117.35	340.57 ± 4.87	3.41 ^a^ ± 0.05	301.28 ± 3.54	3.00 ^Aa^ ± 0.02
	CT (266)	10,253.00 ± 69.97	336.61 ± 3.05	3.30 ^b^ ± 0.03	303.40 ± 2.22	2.96 ^b^ ± 0.01
	TT (631)	10,267.00 ± 60.75	339.68 ± 2.72	3.33 ^ab^ ± 0.03	302.52 ± 1.97	2.96 ^Bb^ ± 0.01
	P	0.17	0.35	**0.03**	0.76	**0.01**
	PVR (%)	0.37	0.06	1.15	0.03	0.86
rs442384907	CC (31)	9739.25 ^B^ ± 143.61	327.68 ± 5.90	3.41 ± 0.06	292.94 ^b^ ± 4.30	3.02 ^A^ ± 0.02
	CT (257)	10,256 ^A^ ± 70.67	336.01 ± 3.08	3.29 ± 0.03	303.26 ^a^ ± 2.24	2.96 ^B^ ± 0.01
	TT (659)	10,282 ^A^ ± 60.45	340.85 ± 2.71	3.34 ± 0.03	303.17 ^a^ ± 1.97	2.96 ^B^ ± 0.01
	P	**6.00 × 10^−4^**	**0.02**	**0.02**	**0.04**	**0.01**
	PVR (%)	2.40	0.44	1.21	0.87	1.78
rs135906064	CC (50)	10,058.00 ± 117.37	340.61 ± 4.87	3.41 ^a^ ± 0.05	301.31 ± 3.54	3.00 ^Aa^ ± 0.02
	GC (267)	10,258.00 ± 69.96	336.88 ± 3.05	3.30 ^b^ ± 0.03	303.54 ± 2.22	2.96 ^b^ ± 0.01
	GG (630)	10,265.00 ± 60.751	339.56 ± 2.72	3.33 ^ab^ ± 0.03	302.46 ± 1.97	2.96 ^Bb^ ± 0.01
	P	0.17	0.44	**0.04**	0.71	**0.01**
	PVR (%)	0.38	0.05	1.13	0.03	0.86
rs136643841	AA (50)	10,059.00 ± 117.37	340.65 ± 4.87	3.41 ^a^ ± 0.05	301.35 ± 3.54	3.00 ^Aa^ ± 0.02
	AG (266)	10,264.00 ± 70.02	337.09 ± 3.06	3.30 ^b^ ± 0.03	303.74 ± 2.22	2.96 ^b^ ± 0.01
	GG (631)	10,262.00 ± 60.73	339.46 ± 2.72	3.33 ^ab^ ± 0.03	302.37 ± 1.97	2.96 ^Bb^ ± 0.01
	P	0.17	0.51	**0.04**	0.63	**0.01**
	PVR (%)	0.39	0.05	1.13	0.04	0.86
rs42427809	CC (183)	10,188.00 ± 76.86	339.39 ^b^ ± 3.30	3.35 ^ab^ ± 0.03	302.46 ± 2.40	2.98 ± 0.01
	CT (460)	10,252.00 ± 63.05	336.24 ^b^ ± 2.80	3.30 ^b^ ± 0.03	302.56 ± 2.03	2.96 ± 0.01
	TT (304)	10,285.00 ± 68.65	342.44 ^a^ ± 3.01	3.36 ^a^ ± 0.03	303.01 ± 2.19	2.96 ± 0.01
	P	0.42	**0.02**	**0.01**	0.96	0.07
	PVR (%)	0.06	0.01	5.03 × 10^−3^	1.35 × 10^−3^	0.19
rs134989007	AA (104)	10,243.00 ± 91.10	344.38 ^a^ ± 3.85	3.38 ^a^ ± 0.04	304.81 ± 2.80	2.99 ^a^ ± 0.01
	AG (371)	10,251.00 ± 66.27	335.00 ^b^ ± 2.92	3.29 ^b^ ± 0.03	301.89 ± 2.12	2.95 ^b^ ± 0.01
	GG (472)	10,250.00 ± 62.59	340.28a ± 2.781	3.35 ^a^ ± 0.03	302.76 ± 2.02	2.96 ^ab^ ± 0.01
	P	0.99	**4.61 × 10^−3^**	**2.10 × 10^−3^**	0.47	**0.03**
	PVR (%)	5.96 × 10^−4^	0.27	0.54	0.06	0.37
rs109837561	CC (104)	10,243.00 ± 91.10	344.38 ^a^ ± 3.85	3.38 ^a^ ± 0.04	304.81 ± 2.80	2.99 ^a^ ± 0.01
	CT (371)	10,251.00 ± 66.27	335.00 ^b^ ± 2.92	3.29 ^b^ ± 0.03	301.89 ± 2.12	2.95 ^b^ ± 0.01
	TT (472)	10,250.00 ± 62.54	340.28 ^a^ ± 2.78	3.35 ^a^ ± 0.03	302.76 ± 2.02	2.96 ^ab^ ± 0.01
	P	0.99	**4.61 × 10^−3^**	**2.10 × 10^−3^**	0.47	**0.03**
	PVR (%)	5.96 × 10^−4^	0.27	0.54	0.06	0.37
rs136361519	CC (471)	10,248.00 ± 62.56	340.2 ^ab^ ± 2.78	3.35 ^a^ ± 0.03	302.69 ± 2.02	2.96 ^ab^ ± 0.01
	CT (372)	10,254.00 ± 66.21	335.12 ^b^ ± 2.92	3.29 ^b^ ± 0.03	301.99 ± 2.12	2.95 ^b^ ± 0.01
	TT (104)	10,243.00 ± 91.09	344.42 ^a^ ± 3.85	3.38 ^a^ ± 0.04	304.83 ± 2.80	2.99 ^a^ ± 0.01
	P	0.99	**0.01**	**2.22 × 10^−3^**	0.50	**0.03**
	PVR (%)	6.00 × 10^−4^	0.27	0.54	0.06	0.37
rs42427812	AA (347)	10,264.00 ± 66.29	341.28 ± 2.92	3.35 ^a^ ± 0.03	302.67 ± 2.12	2.96 ± 0.01
	AG (438)	10,262.00 ± 63.74	336.13 ± 2.83	3.30 ^b^ ± 0.03	302.76 ± 2.05	2.96 ± 0.01
	GG (162)	10,189.00 ± 79.46	340.4 ± 3.40	3.36 ^a^ ± 0.03	302.55 ± 2.47	2.98 ± 0.01
	P	0.54	**0.04**	**0.01**	0.99	0.11
	PVR (%)	0.04	2.95 × 10^−3^	0.07	1.74 × 10^−4^	0.18
rs137589167	AA (471)	10,248.00 ± 62.56	340.20 ^ab^ ± 2.78	3.35 ^a^ ± 0.03	302.69 ± 2.02	2.96 ^ab^ ± 0.01
	AG (372)	10,254.00 ± 66.21	335.12 ^b^ ± 2.92	3.29 ^b^ ± 0.03	301.99 ± 2.12	2.95 ^b^ ± 0.01
	GG (104)	10,243.00 ± 91.09	344.42 ^a^ ± 3.85	3.38 ^a^ ± 0.04	304.83 ± 2.80	2.99 ^a^ ± 0.01
	P	0.99	**5.77 × 10^−3^**	**2.23 × 10^−3^**	0.50	**0.03**
	PVR (%)	6.00 × 10^−4^	0.27	0.54	0.06	0.37
rs42427748	AA (169)	10,186.00 ± 78.27	340.85 ^B^ ± 3.36	3.37 ^a^ ± 0.03	302.73 ± 2.44	2.98 ^a^ ± 0.01
	AG (439)	10,252.00 ± 63.29	335.36 ^B^ ± 2.81	3.29 ^Bb^ ± 0.3	302.39 ± 2.04	2.96 ^b^ ± 0.01
	GG (339)	10,280.00 ± 67.18	342.49 ^A^ ± 2.95	3.36 ^Aa^ ± 0.03	303.05 ± 2.15	2.96 ^b^ ± 0.01
	P	0.44	**3.50 × 10^−3^**	**1.00 × 10^−3^**	0.92	**0.03**
	PVR (%)	0.06	1.25 × 10^−3^	0.07	1.07 × 10^−4^	0.27
rs135517857	AA (338)	10,276.00 ± 67.21	342.39 ^A^ ± 2.96	3.36 ^Aa^ ± 0.03	302.89 ± 2.15	2.96 ^b^ ± 0.01
	AG (439)	10,257.00 ± 63.31	335.48 ^B^ ± 2.81	3.29 ^Bb^ ± 0.03	302.58 ± 2.04	2.96 ^b^ ± 0.01
	GG (170)	10,182.00 ± 78.13	340.67 ^B^ ± 3.35	3.37 ^a^ ± 0.03	302.57 ± 2.44	2.98 ^a^ ± 0.01
	P	0.42	**0.01**	**9.78 × 10^−4^**	0.98	**0.03**
	PVR (%)	0.04	0.05	0.03	7.27 × 10^−4^	0.14
rs42427776	AA (170)	10,182.00 ± 78.13	340.69 ^B^ ± 3.35	3.37 ^a^ ± 0.03	302.58 ± 2.44	2.98 ^a^ ± 0.01
	AG (440)	10,259.00 ± 63.28	335.57 ^B^ ± 2.81	3.29 ^Bb^ ± 0.03	302.65 ± 2.04	2.96 ^b^ ± 0.01
	GG (337)	10,273.00 ± 67.26	342.29 ^A^ ± 2.96	3.36 ^Aa^ ± 0.03	302.79 ± 2.15	2.96 ^b^ ± 0.01
	P	0.43	**0.01**	**1.00 × 10^−3^**	0.99	**0.03**
	PVR (%)	0.06	7.48 × 10^−4^	0.07	2.16 × 10^−4^	0.26
rs42427803	CC (170)	10,198.00 ± 78.18	340.50 ^B^ ± 3.35	3.36a ± 0.03	302.96 ± 2.44	2.98 ^a^ ± 0.01
	CT (438)	10,248.00 ± 63.30	335.48 ^B^ ± 2.81	3.29 ^Bb^ ± 0.03	302.31 ± 2.04	2.96 ^ab^ ± 0.01
	TT (339)	10,280.00 ± 67.18	342.50 ^A^ ± 2.95	3.36 ^Aa^ ± 0.03	303.04 ± 2.15	2.96 ^b^ ± 0.01
	P	0.54	**4.82 × 10^−3^**	**2.51 × 10^−3^**	0.89	**0.04**
	PVR (%)	0.04	6.79 × 10^−5^	0.04	1.34 × 10^−4^	0.24
rs132741637	CC (472)	10,250.00 ± 62.53	340.28 ^a^ ± 2.78	3.35 ^a^ ± 0.03	302.76 ± 2.03	2.96 ^ab^ ± 0.01
	CT (371)	10,251.00 ± 66.26	335.00 ^b^ ± 2.92	3.29 ^b^ ± 0.03	301.89 ± 2.12	2.95 ^b^ ± 0.01
	TT (104)	10,243.00 ± 91.09	344.38 ^a^ ± 3.85	3.38 ^a^ ± 0.04	304.81 ± 2.80	2.99 ^a^ ± 0.01
	P	0.99	**4.66 × 10^−3^**	**2.10 × 10^−3^**	0.47	**0.03**
	PVR (%)	5.96 × 10^−4^	0.27	0.54	0.06	0.37

Note: The number in the table represents the least squares mean (LSM) + standard error (SE); the *p* value represents the significance for the genetic effects of SNPs; the significant *p* values (*p* < 0.05) are bolded; a and b within the same column with different superscripts mean *p* < 0.05; A and B within the same column with different superscripts mean *p* < 0.01; and the PVR represents the phenotypic variance ratio.

**Table 2 ijms-26-02255-t002:** Genetic associations of haplotype combinations of the COL6A1 gene with milk production traits in Chinese Holstein cows (LSM + SE).

Haplotype Combination (NO.)	Milk Yield (kg)	Fat Yield (kg)	Fat Percentage (%)	Protein Yield (kg)	Protein Percentage (%)
H1H1 (52)	10393.00 ^ab^ ± 128.23	347.54 ^ABab^ ± 5.37	3.37 ± 0.05	306.62 ^abd^ ± 3.91	2.96 ± 0.03
H1H2 (65)	10076.00 ^ACa^ ± 114.01	343.65 ^Aac^ ± 4.82	3.44 ± 0.04	305.96 ^ABacd^ ± 3.50	3.04 ± 0.03
H1H3 (56)	10061.00 ^ACa^ ± 119.89	333.26 ^ADacd^ ± 5.03	3.33 ± 0.04	303.76 ^ACad^ ± 3.67	3.02 ± 0.03
H1H4 (57)	9944.02 ^Aa^ ± 122.12	325.41 ^Ac^ ± 5.12	3.29 ± 0.04	297.64 ^Dd^ ± 3.73	2.99 ± 0.03
H2H2 (43)	10800.00 ^Bb^ ± 133.69	368.27 ^Bb^ ± 5.56	3.44 ± 0.05	322.24 ^Bb^ ± 4.06	2.99 ± 0.03
H2H3 (55)	10429.00 ^ab^ ± 122.61	336.86 ^ACDa^ ± 5.15	3.26 ± 0.05	309.15 ^abd^ ± 3.75	2.97 ± 0.03
H2H4 (60)	10548.00 ^BCb^ ± 119.36	356.54 ^BDbd^ ± 5.02	3.4 ± 0.04	318.66 ^BCbc^ ± 3.66	3.02 ± 0.03
H2H5 (49)	10451.00 ^ab^ ± 128.32	348.18 ^ABab^ ± 5.35	3.35 ± 0.05	314.82 ^ABab^ ± 3.90	3.02 ± 0.03
*p*	**<1.00 × 10^−4^**	**<1.00 × 10^−4^**	**0.03**	**<1.00 × 10^−4^**	0.23

Note: The number in the table represents the least squares mean (LSM) + standard error (SE); H means haplotype; H1: TCCGAACTTGGTGTCAAGAGTC; H2: TTGGAATTTGGTGTCAAGAGTC; H3: CCCAGGTCCCACACTGGAGACT; H4: TCCGAATTTGGTGTCAAGAGTC; H5: TCCGAATTTGGCACTGGAGACT (the underlined SNP represents that the locus in this haplotype is different from the locus in several other haplotypes); the number in the bracket represents the number of cows for the haplotype combination; *p* value shows the significance for the genetic effects of haplotype combination; a, b, c, and d within the same column with different superscripts mean *p* < 0.05; and A, B, C, and D within the same column with different superscripts mean *p* < 0.01.

## Data Availability

The datasets supporting the conclusions of this article are available upon request from the corresponding author.

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
