# Peer review of "COL6A1 Promotes Milk Production and Fat Synthesis Through the PI3K-Akt/Insulin/AMPK/PPAR Signaling Pathways in Dairy Cattle"

_ijms, 2025, doi:10.3390/ijms26052255_

Round 1
Reviewer 1 Report
Comments and Suggestions for Authors
Overall, the study described in the publication is well-designed and the results are presented in a clear manner. Thus, not only is an in-depth study of QTL in the cow genome provided, but also an explanation of the function of the gene/protein studied is found.
The text is generally well-written, although there are places where more explanations of abbreviations would be needed.
Does the fact that the numbering of Figures and Tables throughout the text is in bold have any special significance, or is it something left out in the text preparation process?
Title:
The title reflects the study described in the publication and provides insight into the final result of the study.
Introduction:
The introduction is relatively short, but it shows why it is necessary and why such a study was conducted.
Results:
Given that the results precede the Materials and Methods, the numbering of the Supplementary tables should also follow the order of the text. This numbering does not hinder the comprehension of the text but can create a slight volume because when reading that there is already a table S3, the reader wonders whether he has missed S1 and S2.
It is also essential to consider which items are first mentioned in the results but only explained in the Materials and Methods. For example, in line 84, the FarmGTEx database is first mentioned, but the link is only given at the end of the article.
Figure 3: The image attempts to assemble all the results from one experiment, ultimately creating an image that can be viewed and studied only when enlarged to 200% in the pdf file. Dividing the images into parts would make inserting larger and more precise images possible. It would also be possible to insert each part after it is mentioned in the text rather than at the end of the subsection, forcing the reader to scroll through it.
The same about Figure 4.
In 101.row Figure 1c? It is need to bee Figure 3c?
I would recommend creating a Supplementary table that decodes the genes mentioned in subsection 2.5, so that the reader does not have to go to the database to decode them. We understand that it would be illogical to put a decoding for each gene in the text, as it would create an incomprehensible text, but adding a reference to the table would be possible.
Accordingly, a reference to the Supplementary Table could also be added to Figures 4f and 4g.
Figure 5 is very beautifully created with a good explanation. A reference to a table where abbreviations are deciphered would also be useful here.
In Table 1, horizontal lines would be desirable to separate the data/results for each SNP, as this type of table is large and requires divisions. Also, two decimal places are sufficient for P and PVR values. I would recommend highlighting the significant P values ​​- by underlining or in bold.
In Table S5, or more precisely in the excel file Table S5, it would be desirable to use the SNP rs number in the Effect analyze part so that the results are consistent with the part of the text where the rs identification is mentioned.
Another thing, if the text mentions Table S5, then it should be mentioned which of the three parts exactly to look at.
In Table S5, in the Haplotype analysis section, where each haplotype is mentioned, it is recommended to highlight (bold, underline or other) the common or rare allele of each SNP or locus, thus indicating additional information about the haplotypes.
The article does not contain Figure 6.
Subsection 2.7 continues to use abbreviations for genes/proteins/transcription factors that are not explained in the text. Accordingly, a reference to the explanation of the abbreviations is required.
Overall, the results section shows very good results, which are arranged in a very clear and logical layout.
Discussion:
The discussion generally explains the results well. However, there are a couple of things that are missing:
1. The results summarize data on the expression level of the COL6A1 gene in various tissues, which is taken from the database, but does not (1) compare the obtained data with, for example, the expression summary in the NCBI database (https://www.ncbi.nlm.nih.gov/gene/511422)
2. It also discusses possible shortcomings of the study, but there is no discussion of what the results can provide in practice. What is the practical application of the results obtained?
Materials and methods:
The methodology is described very well and precisely, providing a clear insight into what was done in the study and an opportunity to repeat the study. Also, the methodology described in this way allows other researchers to draw ideas on improving their research.
Reviewer 2 Report
Comments and Suggestions for Authors
Dear Authors,
Please rewrite carefully some expressions used in manuscript e.g. in abstract you wrote that "Previously, we discovered a functional gene involved in milk fat metabolism, collagen type VI alpha 1 (COL6A1), from liver transcriptome data across various lactation periods of cows."
with all respect but you did not discovered COL6A1 gene - you only performed analysis (both molecular and statistical) that proved assocation of this gene with in this case milk traits.
-line 34 - what "Breeding theories" authors are talkign about? - or just different "breeding" or "selection methods"
-lines 58-59 - COL6A1 is a gene or superfamily? you wrote that "COL6A1 is a protein superfamily" which is not true - collagenes are superfamily of proteins involved in integrity and COL6A1 is one of them
Table 1 heading should be corrected, also there is lack of explanation of statistical symbols
line 166 - for SNP rs210433593 there is only association with fat yield - so why authors wrote that SNP was associated with all five traits?
there is only 10 SNPs showing statistical differences - (they are marked by letters "a" and "b") why authors wrote about 15 SNPs in line 165? e.g. rs135862366 SNP according to data in table 1 there is no statistical differences between genotypes. And according to what data you included in Table 1 there should be written that there are differneces between genotypes for milk production traits not that were significantly associated with milk production traits
line 268 - how samples were collected from animals? were they sacrificed? where is permission of Local Ethical/Animal Care commision?
as i see in model and it explanation (lines 404-410) authors were colelcting data in different years and periods of year. It must be added adn clearly desribed in Methods section. You used data from 947 cows in period of 4 years (2012-2015). what is cow were in herd e.g. in 2012 and 2013 and were evaluated for milk yield? Do you used for one cow only data from one year or from more years and how it was analysed?
Comments on the Quality of English Language
As i wrote there are many wrongly used words liek "Previously, we discovered a functional gene involved in milk fat metabolism". Again authors did not discovered this gene they may find association with this gene
Round 2
Reviewer 2 Report
Comments and Suggestions for Authors
Authors should use in statistical analysis descripsion only '" statisical association" or "lack of association". Missleading is using "strong assocaition" etc. In 174-193 we have 3 "types" of association: strong association, statistical association and associacion.
In Tables where authors show values, numbers after decimal point should be always same
